

# Pyroclastic flow mitigation strategies: a new perspective for the red area

Mauro Iacuaniello[1], Andrea Montanino[1], Daniela De Gregorio[2], Giulio Zuccaro[1,2]

[1]Department of Structures for Engineering and Architecture, University of Naples, Naples, Italy

[2] PLINIVS - LUPT Study Centre, University of Naples Federico II, Via Toledo 402, 80134, Naples, Italy

*Correspondence to* Mauro Iacuaniello (mauro.iacuaniello@unina.it)

**Abstract.**

This paper intends to show basic strategies and technical solutions that may reduce the consequence of a volcanic eruption on buildings. One of the most dangerous eruptive phenomena is pyroclastic flow, a gas-solid mixture which can flow slope

down up to reach considerable distances from the point of emission, with a speed that can easily exceed 100km/h (~ 30m/s). The damage caused by the impact on buildings depends on the combination of several factors: the duration of phenomenon, the temperature of flow and the pressure produced by the impact. The study tries to define a design-oriented approach of retrofitting and new construction with specific reference to the possible eruption of Vesuvio or Campi Flegrei in the Campania Region of Italy. For both red areas, which are directly affected by the flows, the Department of Civil Protection

defined only one solution, that is the preventive evacuation. Whilst through the proposed mitigation strategy, the task is to establish a resilient sector within the red areas where buildings can withstand pressure and temperature in order to achieve both economic benefits and make reconstruction easier.

**Keywords:** Mitigation strategies; Volcanic risk; Pyroclastic flows.

## 1.Introduction

Explosive volcanic eruptions have the potential to cause casualties and economic losses, unless appropriate mitigation measures can be taken. Inside the metropolitan area surrounding the city of Naples (Campania Region, Italy) there are two active volcanic systems, the Somma-Vesuvius and the Campi Flegrei, whose geological history allows to believe that they

will be able to produce, in the future, some explosive eruptions. The hazard of both volcanoes, the high exposed value of urban area, which counts about three thousand people, and the high vulnerability of the urban settlements make the Neapolitan territory one of the riskiest volcanic area in the word (Zuccaro and De Gregorio, 2011). Volcanic eruptions encompass different hazards as volcanic earthquake, ash fall, pyroclastic flows and subsequent floods and mudflows. The most dangerous phenomenon is the pyroclastic flows, which can occur with little warning, move at high speeds, and have

enormous destructive power (Spence et al., 2010). Although the Emergency Plans provide for the preventive evacuation of the areas affected by the phenomenon, the protection of the openings represents the main objective as it happened during the



volcanic eruption on the Caribbean island of Montserrat (British Overseas Territory), the openings and especially the unprotected ones were the first elements of the building envelope to be compromised and causing considerable damage to the structure itself. Therefore, in this paper, the vulnerability analysis of these elements present in the Neapolitan areas and

the analysis of mitigation strategies valid for both pressures and temperatures have been carried out first. Through these analyses, it is possible to define within the red areas possible resilient sectors in which the recovery of buildings could have both economic and technical feasibility advantages.

## 2. Volcanic Phenomena

Explosive eruptions, foreseen in Vesuvius and Campi Flegrei regions, are characterized by a powerful ejection of pyroclastic material producing the eruptive column composed of gas-solid dispersal, which rises vertically from the eruptive centre and extending alongside the wind action, which disperses the haze according to its own direction. Hence, part of clasts falls from the column into gas-solid dispersals designated as pyroclastic flows and surges, respectively, with a high or low density of particles; the rest of the clasts descends via gravity (ash fall deposits) or is exploded directly in the air from the crater

(ballistics). Furthermore, the eruption process involves seismic shakings plus trembles, (produced by the cracking of rocks compressed by the magma motion); landslides, floods and lahars (flows of water mixed to pyroclastic material generated by heavy rains that follow the eruptions); and tsunami (induced by submarine earthquakes or huge pyroclastic flows which arrive in the sea). (Zuccaro and De Gregorio, 2013)

Particularly, for the Campi Flegrei, unlike what happens in volcanoes with central apparatus, such as Vesuvius, the area of

the possible opening of the eruptive vent is very large; moreover, regarding fallout, it should be considered that, unlike Vesuvius, the city of Naples is downwind of the dominant winds and would therefore be involved. While the hypothesized scenario for Vesuvius is a Sub-Plinian event, corresponding to a volcanic explosivity index VEI =4), with a conditioned probability of occurrence rather less than 30% (Marzocchi et al., 2004). On the other hand, for Campi Flegrei, as eruption scenario, a medium-size event was assumed to occur with a conditioned probability of occurrence rather than about 24%

(Costa et al., 2009). In this paper, the reference eruption is a sub-Plinian I type (SPI) event, which produces all the possible destructive phenomena associated with a violent explosive event: precursory earthquakes, ashfall (fallout), pyroclastic flows (PDC = Pyroclastic Density Current), ballistics, lahars, floods, landslides and tsunami. (Neri et al., 2008b)

The pyroclastic flows are the most dangerous volcanic phenomenon produced by a sub-Plinian eruption. They are generated by the gravitational collapse of the eruptive column at the end of the ash fall phase. The pyroclastic flows are a suspension of

gas and solid particles of various sizes. Their hazard at Vesuvius had been studied by numerical modelling by Todesco et al. (2002) and Esposti Ongaro et al. (2002). In structural assesses in order to evaluate the vulnerability, the action due to pyroclastic flows could be considered as a uniformly distributed static pressure (Petrazzuoli and Zuccaro, 2004), within a temperature range between 200 and 350 °C (Gurioli et al. 2008). In previous studies, (Esposti Ongaro et al. 2008; Neri et al.





2007), through a 4D model, where the Vesuvius was schematized with its real geometric dimension and the variable time is included. After 900 s since the pyroclastic flow's origin, a pressure of 1-3 kPa at 7.5 km from the vent, with a temperature of 250 °C was assessed.

The experience from the 1997 Montserrat eruption (UK Caribbean Islands) had highlighted that a building could withstand moderate pyroclastic flows pressure (1–5 kPa), whereas if one or more openings (windows and doors) collapse, leaving hot gas and ash to go inside and the entire building is likely to be destroyed (Baxter et al. 2005). Indeed, the contents of the construction and any timber structure are likely to catch fire; at the same time, the principal structural walls and roofing will suffer a combination of internal and external pressures, which will cause partial or total failure (Spence et al. 2004a). In general, the first elements to fail are the glasses and the shutters. Nonetheless, the lateral resistance of a building to pyroclastic flow strongly depends on the design criteria applied to resist ordinary load conditions: of course, an earthquake-resistant building has larger strength and stiffness capabilities than a non-seismic building. The structural behaviour of buildings under the impact of pyroclastic flows is not akin to that induced by earthquakes, since the horizontal pressure is not a cyclic action. So, the structural response is less influenced by the ductility, like the capability to dissipate energy.

## 3.Exposure

### 3.1 Description of building data of Vesuvius and Campi Flegrei areas

In order to define the best mitigation strategies, it has been necessary to arrange some available data about the buildings in the red zones, gathered up by the P.LIN.V.S. Among these data, the useful for the purpose are the vertical structure and material of windows frame and shutters. Surveys in both areas were carried out to determine diffusion of the different construction types and their features which determine the resistance of the buildings to the impact of pyroclastic flows. There is a wide diffusion of buildings framed in reinforced concrete with thick infills panels and masonry structures with square blocks in brick or tuff (Fig.xx). While, there is a widespread diffusion of aluminium and wood windows with UPVC shutters for both the areas (Fig.1, Fig.2). Furthermore, in this study the most studied openings are the windows, which together with the doors represent a weak point of the building envelope during a pyroclastic flow event, since the dynamic pressure exceeds the characteristic resistance of them, increasing the vulnerability (Spence et al. 2004).

So, the data about openings have been divided into three groups:

- Size of openings;
- Frame types;
- Shutters types.

Each of these characteristics is important in assessing the vulnerability and so in defining the adequate mitigation measures.

Besides, the sizes of openings were recorded in three classes:

- Large windows, whose area is greater than 1,5 $m^2$;





• Typical windows, whose area range from 0.5 to 1.5 m$^2$;

• Small windows, whose area is less than 0.5 m$^2$.

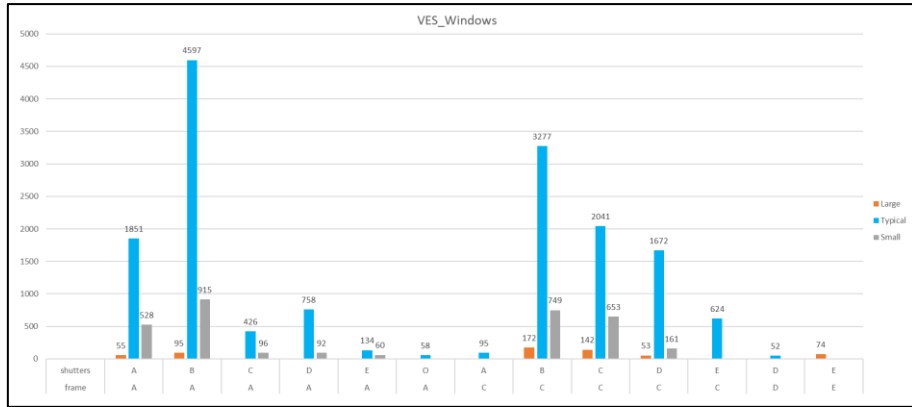

**Figure 1 Number of windows in Vesuvius area divided by size and material**

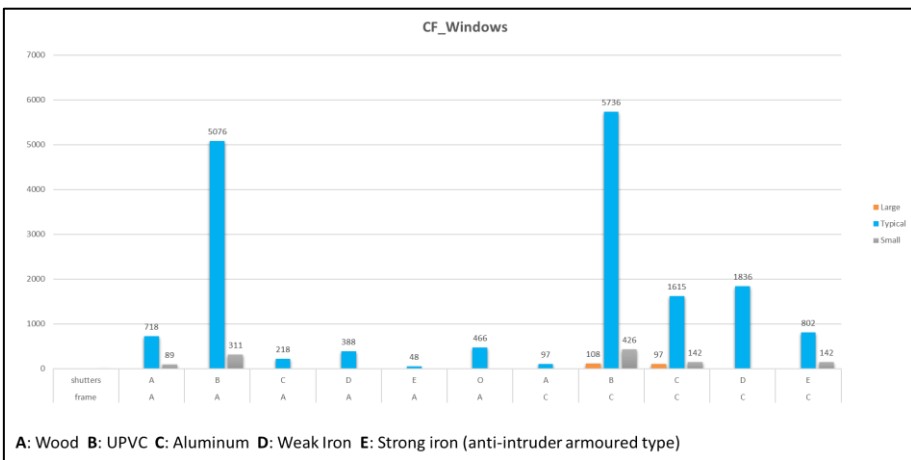

**Figure 2 Number of windows in Campi Flegrei divided by size and material**

### 3.2 Aluminium windows

The EN AW-6060 alloy is the most widespread extrusion alloy on the European market, thanks to its high hot forming speed. The alloy allows the production of profiles with even complex sections, including cavities and multiple grooves, to 105 bring the design of the extruded part as close as possible to that of the finished product and to minimize intermediate machining. The mechanical model used is Ramberg-Osgood (1) and the characteristics of the alloy are (Table 1):






| Table 1 Physical and mechanical properties of aluminium EN-AW 6060 ||
|---|---|
| Density | 2700 [kg/m$^3$] |
| Elastic modulus | 70000 [MPa] |
| Breaking voltage | 160 [MPa] |
| Poisson Coefficient | 0.33 |
| Specific heat capacity | 900 [J/kgK] |
| Thermal conductivity | 238 [W/mK] |
| Thermal expansion | 3,7 e$^{-7}$ [1/K] |

$$\varepsilon = \frac{\sigma}{E} + K \left(\frac{\sigma}{\sigma_y}\right)^{\frac{1}{n}} \tag{1}$$

- $\sigma_y$ is the yield strength of the material,
- $\sigma$ is the value of the stress considered,
- E Young's modulus,
- n exponent of the hardening of the material.

The type of glass commonly used is composed of silica oxide and lime. As defined in the Instructions for the design, execution and control of constructions with structural glass elements, the latter can be considered a homogeneous, isotropic material with linear elastic behaviour at breakage, both tensile and compressive. The characteristics of this type of glass (Table 2) are:

| Table 2 Physical and mechanical properties of soda lime glass ||
|---|---|
| Density | 2400 [kg/m$^3$] |
| Elastic Modulus | 71000 [MPa] |
| Ultimate Tensile Strength | 41 [MPa] |
| Compressive strength | 300 [MPa] |
| Poisson's ratio | 0.33 |
| Specific Heat Capacity | 800 [J/kgK] |
| Thermal Conductivity | 1 [W/mK] |

From the surveys, it is not possible to determine what type of section is used. Therefore both the thermal break window (Fig.3) and the window made entirely of aluminium have been analysed. In addition, the technology hypothesized, as the most common, is that of insulating glass, which indicates the set of two or more sheets of equal or variable thickness, separated by a cavity, usually of air. For the analyses, two panes of the same thickness, i.e. (4/5/6) mm.



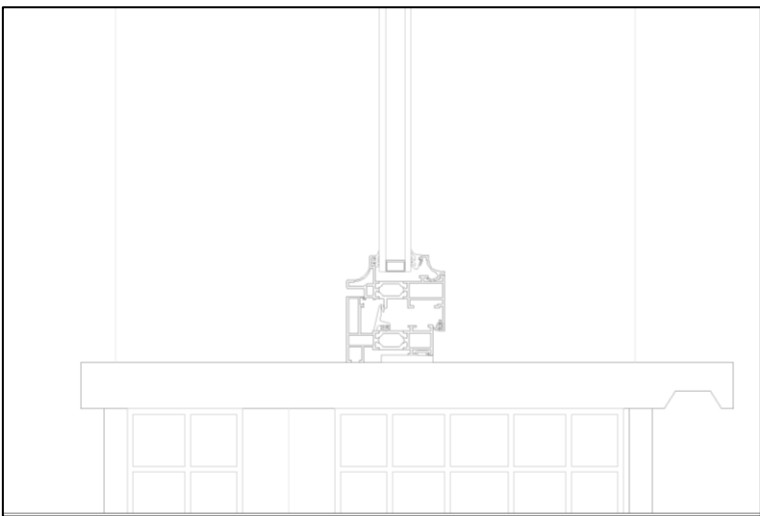

**Figure 3 Thermal break aluminum window section**

## 3.2 Timber Windows

Another type of material widespread in the Phlegrean and Vesuvian area for the construction of windows and doors is wood. The choice of this material depends mainly on the good thermal insulation characteristics compared to UPVC and aluminum windows and doors. In fact, this choice is economically disadvantageous because wood is certainly a more delicate material compared to PVC and aluminum, as it requires regular maintenance, and because the price of wooden windows and doors is still higher than that of aluminum and PVC.

There are several species of wood, belonging to the broadleaf and conifer families, used for the construction of windows and doors:

- chestnut,
- fir,
- pine,
- douglas.

Besides a first hypothesis was to consider the material as a homogeneous and isotropic, whose behavior has been hypothesized linear elastic until breakage. Therefore, the characteristics of the two wood species (Tab. 4, Tab. 5).



| Table 4 Physical and mechanical properties of Pine | |
| --- | --- |
| Density | 532 [kg/m$^3$] |
| Elastic Modulus | 13700 [MPa] |
| Tensile Strength | 85 [MPa] |
| Compressive Strength | 45 [MPa] |
| Poisson's ratio | 0.33 |

| Table 5 Physical and mechanical properties of Chesnut | |
| --- | --- |
| Density | 630[kg/m$^3$] |
| Elastic Modulus | 114000 [MPa] |
| Tensile Strength | 95 [MPa] |
| Compressive Strength | 51 [MPa] |
| Poisson's ratio | 0.30 |


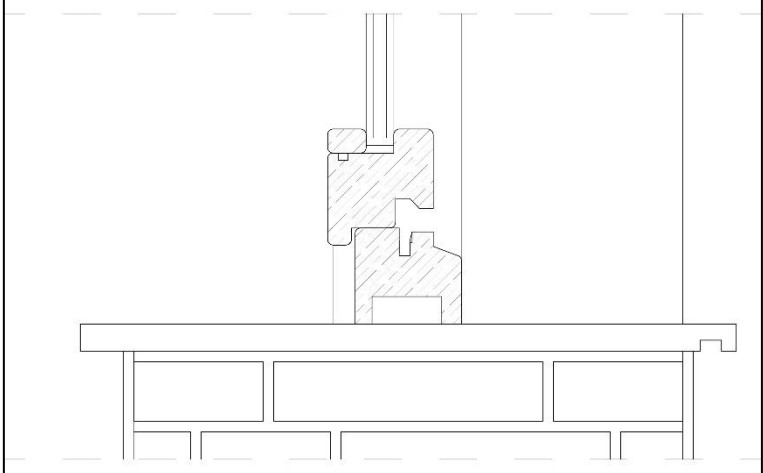

**Figure 4 Wood window section**

### 3.3 UPVC Windows

Another common material to produce windows and doors is UPVC. UPVC is a material composed of macromolecules which are in turn formed by hydrogen, carbon and chlorine atoms. A UPVC window and door frame has very similar characteristics to those of wood. UPVC has a low mechanical resistance; to overcome this, multi-chamber profiles are extruded and reinforced with the help of metal elements (Fig.5). These windows and doors have the main characteristic of resisting very well to the aggressions of atmospheric agents, are very light



and offer good thermal insulation (Mottura and Pennisi, 2014). PVC is a thermoplastic so its mechanical characteristics (Tab.6) are highly dependent on the glass transition temperature ($T_G$); that is the temperature, below which the physical properties of plastics change to those of a glassy or crystalline state. Above $T_g$ they behave like rubbery materials, below the $T_g$ a plastic's molecules have relatively little mobility (Ebnesajjad, 2016). In particular PVC has a glass transition temperature of 80°C. Thus, considering the temperatures expected

in the red areas, PVC windows and doors fail and therefore consider the building totally exposed to the impact of pyroclastic flows.

| Table 6 Physical and mechanical properties of UPVC | |
| --- | --- |
| Density | 1400 [kg/m$^3$] |
| Elastic Modulus | 3700 [MPa] |
| Tensile Strength | 47 [MPa] |
| Coefficient of thermal expansion | 0,8e$^{-4}$ [1/K] |
| Poisson's ratio | 0.40 |

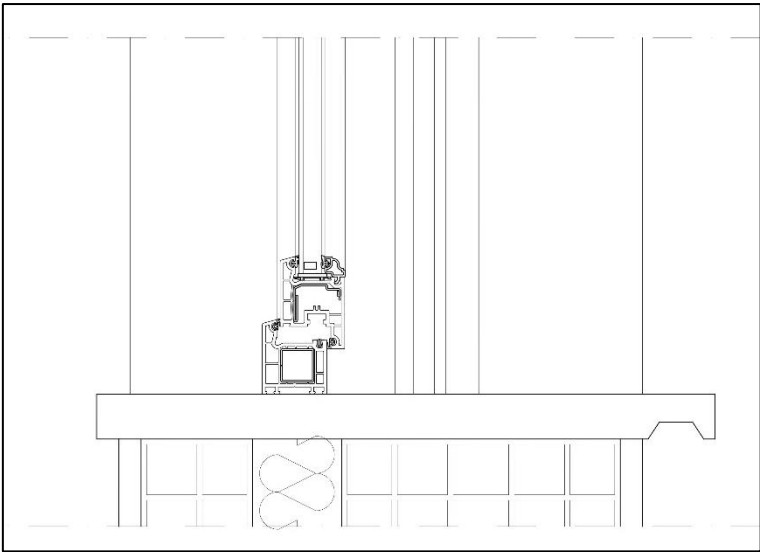

**Figure 5 UPVC window section**

**4 Vulnerability Assessment**

**4.1 Mechanical Assessment**

To assess the vulnerability of the frames to the pressures expected in both areas, a two-dimensional stationary linear static model (2.1) of the physic Mechanical Structures has been set up for both aluminium and wooden frames. Since the resistance of openings to dynamic pressure depends on several factors of which the most important are the size; therefore, three



different heights have been considered for each group of windows, i.e. for the large openings a height of 2.4 m had been considered, for the Typical a height of 1.2 m and for the small a height of 0.8 m has been considered; and for each window size the different thicknesses of the glass have been considered. Furthermore, it has been considered a fixed constraint (2.3) at the base of the wall on which the window is placed (Fig.3). Additionally, it has been considered half section in order to reduce the computational time, assuming a condition of symmetry (2.4) in the upper part of the window.


$$0 = \nabla \cdot (FS)^{T} + F_V \quad (2.1)$$
$$F_V = I + \nabla_u \quad (2.2)$$
$$u = 0 \quad (2.3)$$
$$u \cdot n = 0 \quad (2.4)$$


In addition, a uniformly distributed load applied (2.5) on the external front has been assumed, in favor of opening, which is linearly variable according to a parameter that has been imposed through a range function.

$$S \cdot n = F_A \quad (2.5)$$
$$F_A = \frac{F_L}{d} \quad (2.6)$$

For modelling better, the problem and overcoming the convergence problem, it has been necessary to insert a Stop Function in the Solver Configuration, imposing an if condition for the glass:

comp1.StressMax > 40[MPa]


Subsequently, from these first mechanical analyses, the glass of 4 mm of large dimensions, therefore with height equals to 2.4 m is the most vulnerable because the calculated breaking pressure is equal to 0.6 kPa (Tab.7). This situation is not entirely similar for wooden frames, as the glass of this type is placed inside the frame without the aid of gaskets, so the glass is perfectly embedded in the frame itself (Tab.8). Although the glass in the case of wooden frames may be suitable to 205 withstand the expected pressures, the problem lies in the resistance of the glass to temperature variation.


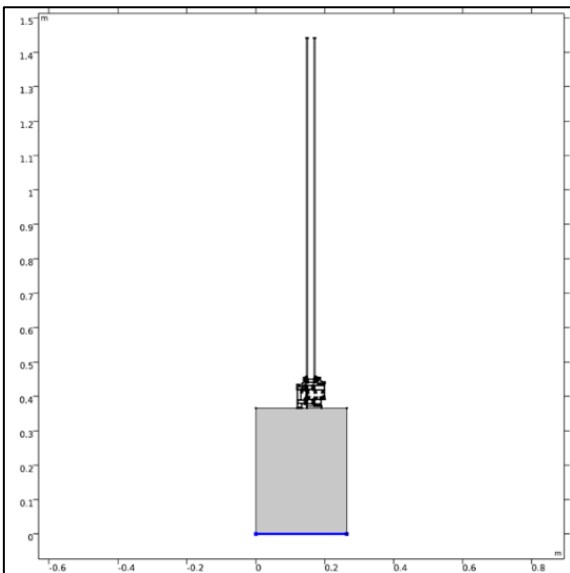

**Figure 6 Fixed constraint condition**

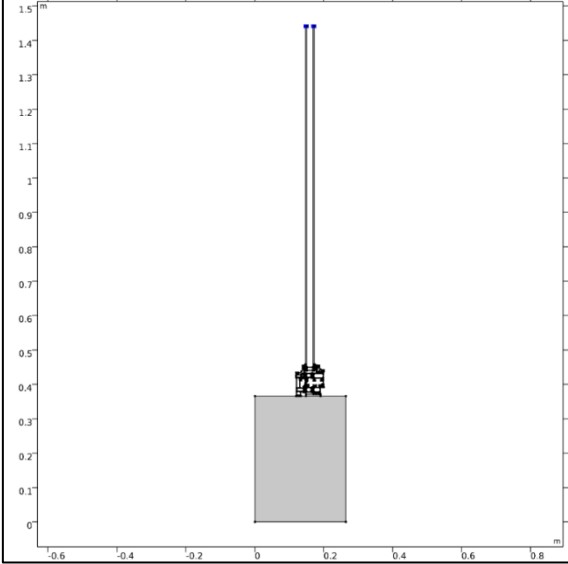

**Figure 7 Symmetry condition**







| Table 7 Glass breakage load aluminium windows | | | | | | | | | |
|---|---|---|---|---|---|---|---|---|---|
| **Frame** | **Glass** | | | | | | | | |
| Aluminium | SODA LIME GLASS | | | | | | | | |
| | Large | | | Typical | | | Small | | |
| | 4mm | 5mm | 6mm | 4mm | 5mm | 6mm | 4mm | 5mm | 6mm |
| | 0,6 kPa | 1 kPa | 1,3 kPa | 2 kPa | 3 kPa | 4,3 kPa | 5,3 kPa | 8 kPa | 10,3 kPa |

| Table 8 Glass breakage load timber windows | | | | | | | | | |
|---|---|---|---|---|---|---|---|---|---|
| **Frame** | **Glass** | | | | | | | | |
| Wood | SODA LIME GLASS | | | | | | | | |
| | Large | | | Typical | | | Small | | |
| | 4mm | 5mm | 6mm | 4mm | 5mm | 6mm | 4mm | 5mm | 6mm |
| | 1.6 kPa | 2.3 kPa | 3 kPa | 4.3 kPa | 6.3 kPa | 9.3 kPa | 8.6 kPa | 14.6 kPa | 21.6 kPa |

### 4.2 Thermal Assessment

Since the pyroclastic flows can be schematised both as pressures and as temperature variations, a thermal analysis also had to be set. The models analysed are aluminium and thermal break aluminium. In order to tackle this problem, two different types of thermal analysis have been accomplished. A 3D heat transfer to assess the thermal shock for the glass and a 2D thermal stress for the entire windows.

### 4.2.1 Thermal Shock

Thermal shock occurs when a thermal gradient causes different parts of an object to expand in different quantities. This differential expansion can also be understood in terms of stress or deformation (3.1). At some point, this stress may exceed the strength of the material, causing a crack to form. If nothing prevents this crack from propagating through the material, the glazing will lose its structural integrity. Glass objects are particularly vulnerable to failure due to thermal shock, due to their low strength and low thermal conductivity. If the glass is then suddenly exposed to extreme heat, the shock will cause 230 the glass to break.

$$\Delta T = \frac{(\sigma_{TS} * (1 - v))}{E * \alpha} \tag{3.1}$$

where:

• $\sigma_{TS}$ is the yield strength of the material,
• $v$ Poisson's ratio,



- E Elastic modulus,
- α coefficient of thermal expansions.

In the case of soda lime glass, the critical temperature is 52 °C. Once the critical temperature is defined, it has been necessary to assess the time to reach it through the heat transfer equation (3.2)

$$\rho C_p \left( \frac{\delta T}{\delta t} + u_{trans} \cdot \nabla T \right) + \nabla \cdot q + q_r = -\alpha T : \frac{dS}{dt} + Q \qquad (3.2)$$

To model this problem properly, a time-dependent study was used in the 60 second interval using the range function (0,1,60) s. Moreover, to model the sudden temperature rise, a ramp function (Fig. 5) was applied on the external face, using the following expression:

$$T = 20[degC] + x * rm1(t[1/s]) \qquad (3.3)$$

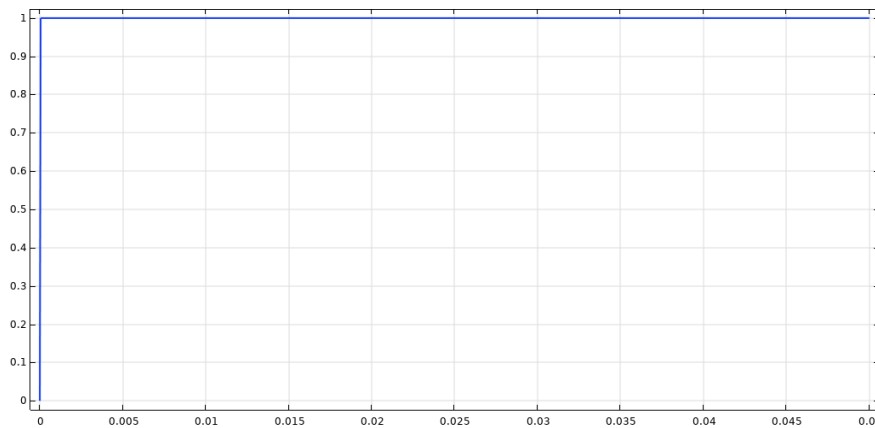

**Figure 8 Ramp Function**

where variable x is equal respectively to 80, 180 and 280 °C so that the glass is subject to three different temperatures: 100°C, 200°C and 300°C.

The results (Fig.xxx) highlight that the soda lime glass reaches the critical temperature around 5 seconds so the common
glass of the windows is totally vulnerable to the temperatures expected in the red areas.



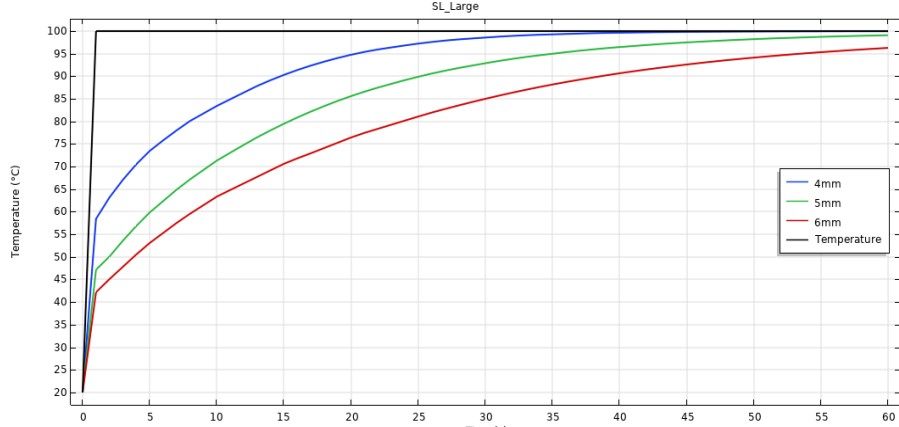

**Figure 9 Reached Temperature for large glass pane**

### 4.2.2 Aluminium Thermal Resistance

To assess the resistance of the aluminium frame, è stato necessario valutare lo stress termico (xxx) due to the different
temperature, which is hypothesized applying on the external front. In these analyses the applied temperature is defined
through two different function:

- a ramp function (3.3) for four different temperatures from 100°C to 400°C, considering as time interval 1200
  seconds, that is the maximum duration of the phenomenon,

- an interpolated function (3.4), that presents a sinusoidal variation of temperatures for 240 seconds, that is the
minimum interval of time (Figure 10). The peculiarity of this function lies in the possibility to reach different
  maximum temperatures by changing the parameter P, indeed if the P is less than one the maximum temperature is
  around 200°C, while if it is greater than 1, the temperature reaches the 500/600 °C. In the first analysis the
  maximum temperature was 400°C.

$$T = T_i + sin(T) * t * P \quad (3.4)$$




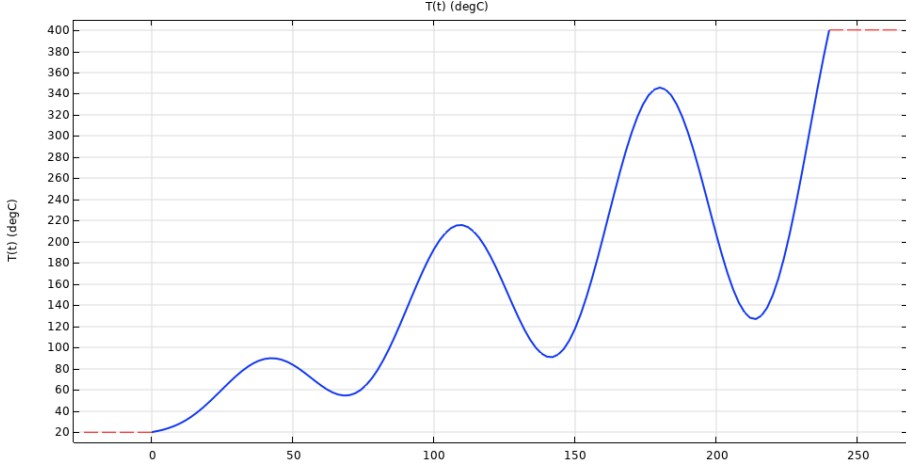

**Figure 10 Interpolation function**

These first analyses show that the aluminium frame can resist the high temperatures, and the real issues are the plastic material of thermal break and sealing. Indeed, from the first analysis, the polyamide reaches this temperature in a range of 600s in the case of 100°C, while in the case of 200°C, 300°C and 400°C the time is reduced to 50s, 20s and 14s respectively

(Fig.11). Besides the EPDM, considering the temperature higher than 100°C, it reaches its critical temperature around the 30 seconds and lower (Fig. 12). Finally, if the temperature is described by the interpolated function (3.4), the results show that also in this case the plastic elements are the most vulnerable components of a window; indeed the polyamide reaches its Tg about 100s (Figure 13). At the same time, the EPDM reaches its Tg around 95 s (Figure 14). Besides the aluminium window has been analysed through the function (3.4) showing a certain resistance to temperature variations, in fact, the frame does

not reach the breaking tension (Fig. 15), and the only vulnerable plastic element is the glass seal.





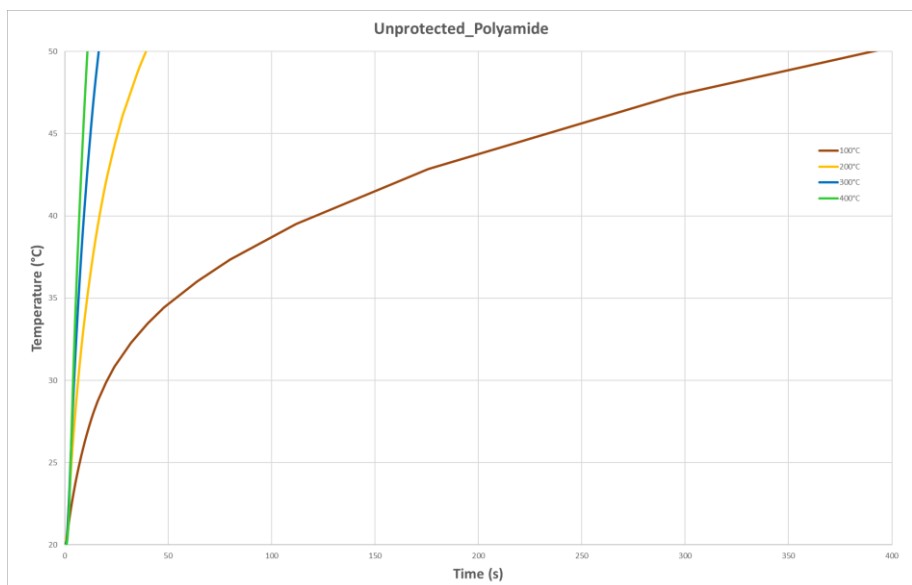

**Figure 11 Temperature of Polyamide of thermal break window**

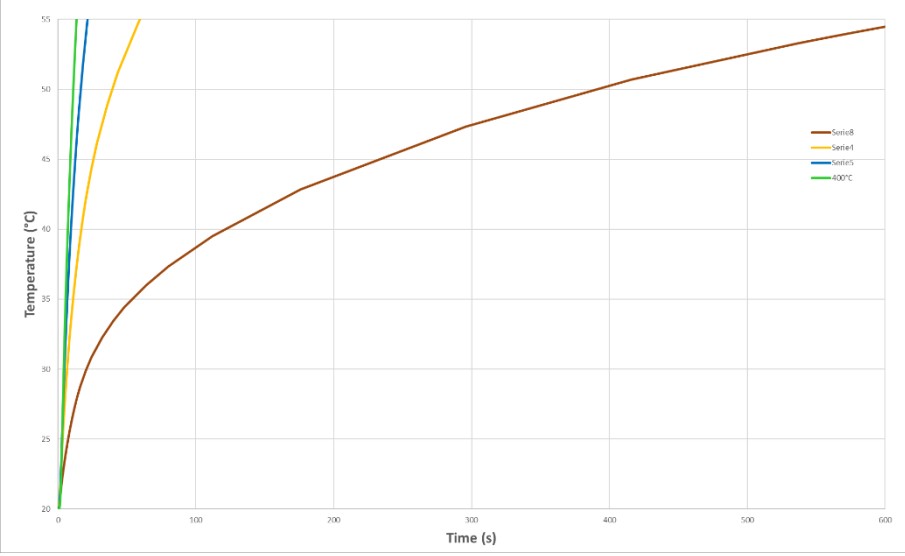

**Figure 12 Temperature of EPDM of thermal break window**






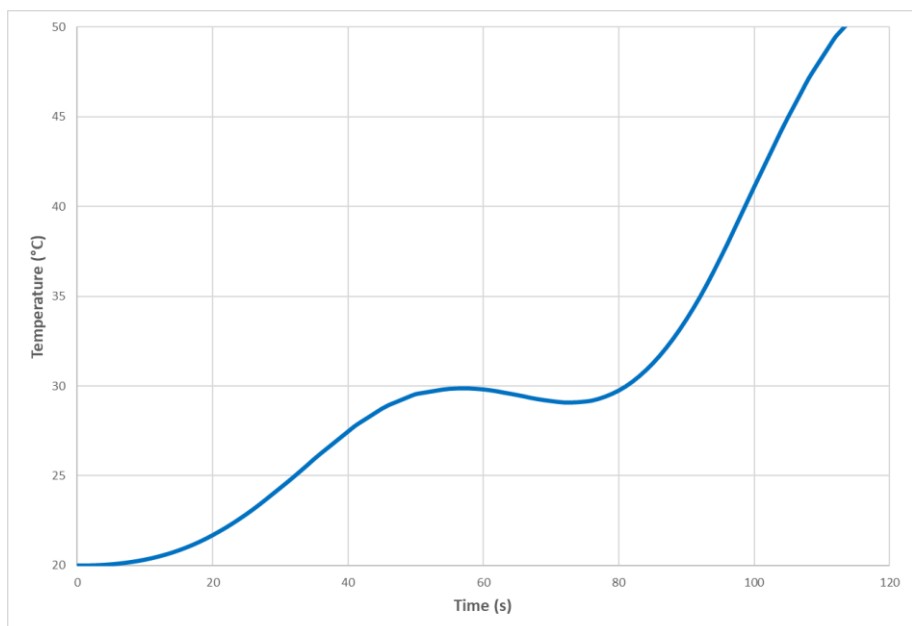

**Figure 13 Temperature of Polyamide of thermal break window**

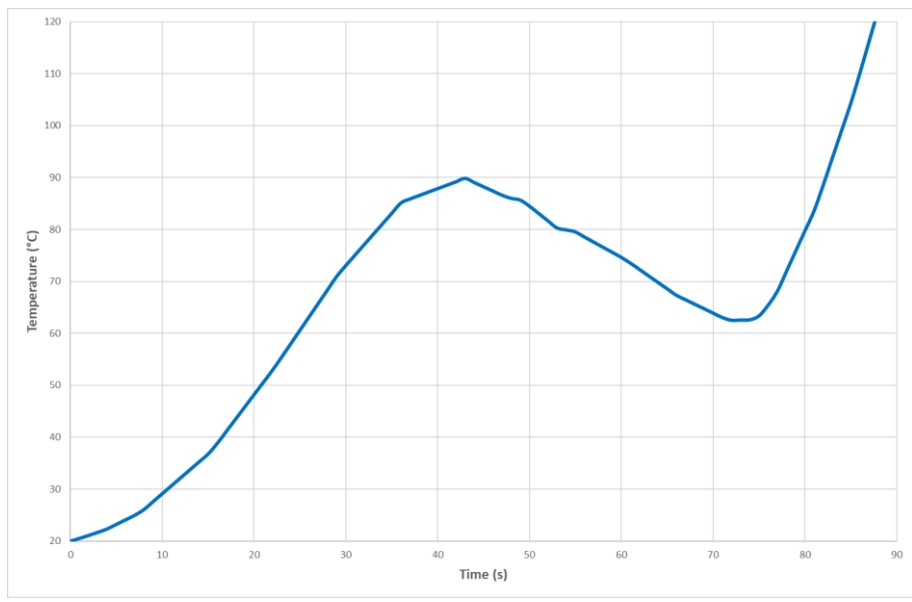

**Figure 14 Temperature of Polyamide of thermal break window**





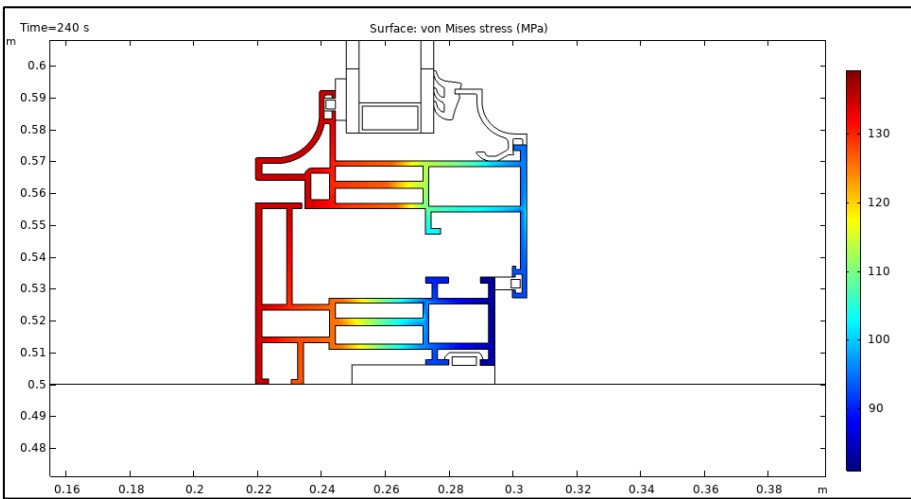

**Figure 15 Thermal stress of aluminium window**

### 5. Mitigation Strategy

The methodology applied to define the correct mitigation strategy is called incremental innovation. This methodology consists in the improvement (or adaptation) of something that already exists. Therefore, interventions on the different components of the element under consideration. In fact, the interventions foreseen for the window consist, first, in replacing the glass with tempered glass, which is a type of safety glass treated with controlled thermal or chemical processes to increase its resistance compared to normal glass. When (3.1) is applied, the critical temperature calculated, considering the characteristics of such glass (Table 9), is 355 °C. Therefore, this option could be suitable for urban settlements present at distances very far from the vent, e.g. in the case of the Vesuvius area it is about 7-8 km. has thermal and mechanical characteristics far better than soda lime glass. In addition, to prevent the plastic elements from reaching the glass transition temperature, a 2 cm wooden element is placed in front of the aluminium frame. (Fig. 16). Furthermore, although the aluminium-only frame has shown a fair resistance to temperature variation, the idea of choosing this model as a starting point for a new mitigation strategy was discarded a priori because the frame does not meet energy-saving standards, so in the case of buildings with this type of frame, it is preferable to replace it.

| Table 9 Physical and mechanical properties of Tempered Glass | |
|---|---|
| Density | 2400 [kg/m$^3$] |
| Elastic Modulus | 70000 [MPa] |
| Tensile Strength | 180 [MPa] |
| Coefficient of thermal expansion | 9 e$^{-6}$ [1/K] |
| Poisson's ratio | 0.23 |





The model, besides, was analysed to the combined actions of pyroclastic flows, in order to fully understand the behaviour. The first step was beginning from the Heat Transfer time dependent study (3.2), considering a maximum temperature of 200°C. After the simulation, the solution is used in the stationary mechanical study (2.1), as initial values of variables solved for and as values of variables not solved for. So, the load, which is defined by a range function, is applied on heated window. Besides for overcoming the convergence problem, it has been necessary, also in this study, to insert a Stop Function in the 310 Solver Configuration, imposing an if condition for the glass:

$$comp1.StressMax > 180[MPa]$$

Thermal analysis shows that the polyamide is totally protected as the maximum temperature recorded is 25°C (Fig.16), so 315 the continuity of the aluminium sections is preserved. In contrast, EPDM is preserved for about 160 seconds before reaching its critical temperature (Fig.17). A possible solution could be the overlapping of an intumescent material which would perform a protective function for this material. Finally, in tempered glass, assuming a thickness of 6mm, there is a maximum tension of about 75 MPa (Fig.18) and therefore still intact. Subsequently, by applying the load (2.6) defined through a range function, the glass is failed for a load equal to 4 kPa (Fig.19).


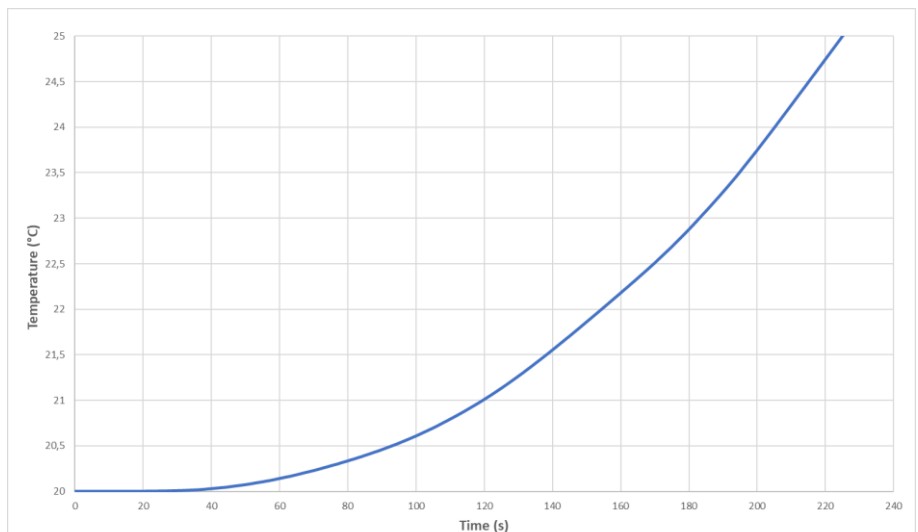

**Figure 16 Reached temperature of polyamide**




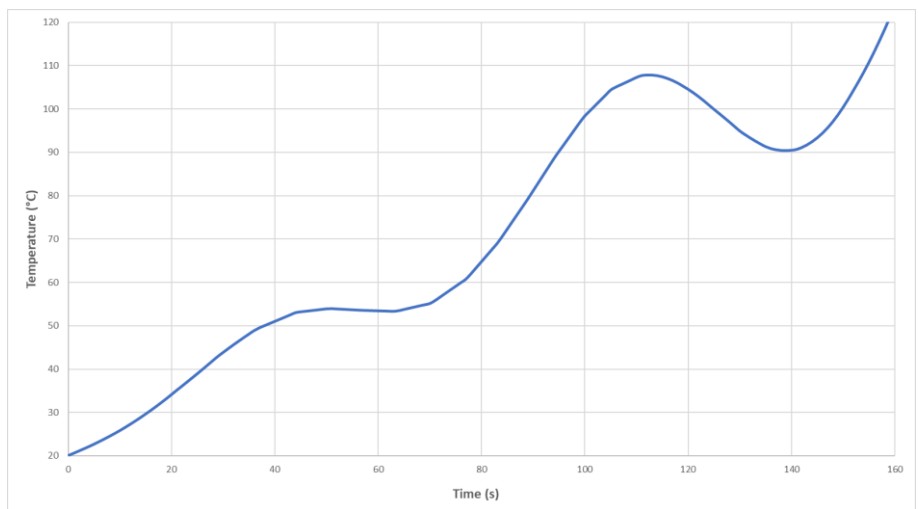

**Figure 17 Reached tempersture of EPDM**

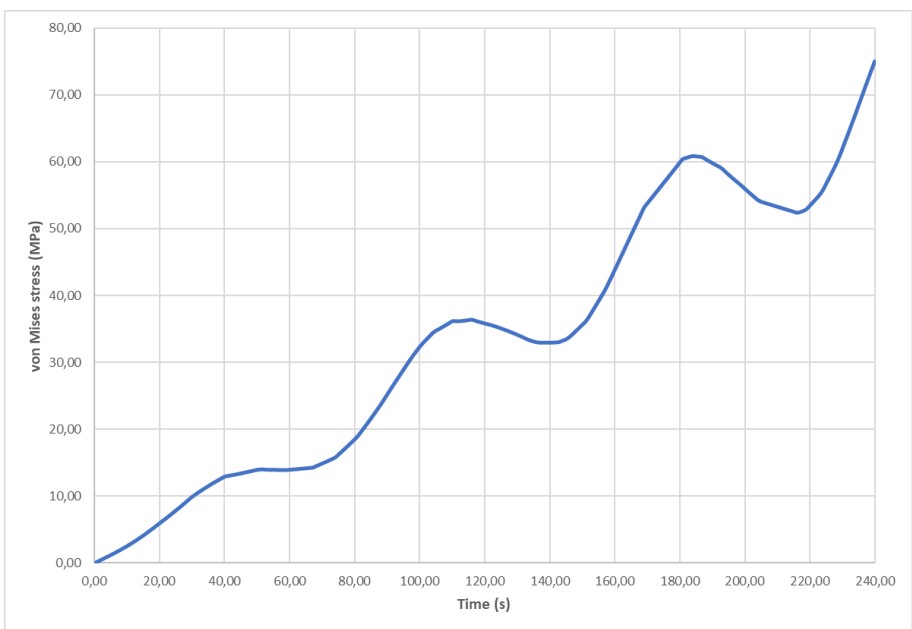

**Figure 18 von Mises stress of aluminium due to the temperature**




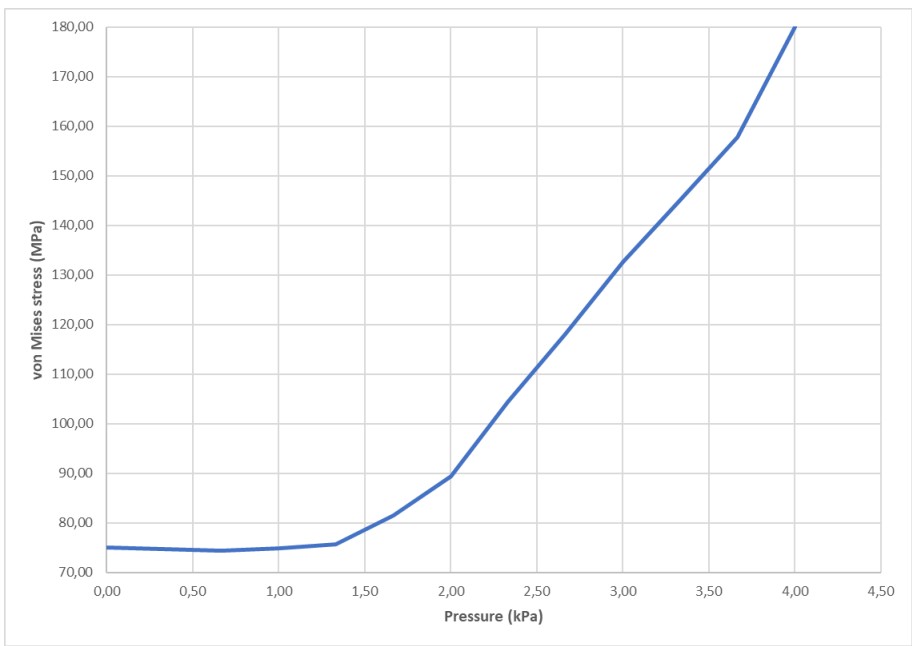


**Figure 19 Breaking load for tempered glass**

**6. Fluid dynamic Assessment**

A fluid-dynamic and heat transfer evaluation have been made in the turbulent field, to study the shielding effect of the
shutter, changing the inclination of the slots and evaluating if the plastic elements reach the relative critical temperatures.
The model uses two different studies: one solving for the turbulent flow around the wall using a Turbulent Flow, k-ε physics
interface (4.1), and the other solving for the heat transfer using a Nonisothermal Heat Transfer physics interface (xxx).

$$\rho(\mathbf{u_{fluid}} \cdot \nabla)\mathbf{u_{fluid}} = \nabla \cdot [-p\mathbf{I} + \mathbf{K}] + \mathbf{F} \quad (4.1)$$


$$p\nabla \cdot \mathbf{u_{fluid}} = 0 \quad (4.2)$$

$$\mathbf{K} = (\mu + \mu_T)(\nabla\,\mathbf{u_{fluid}} + (\nabla_{\mathbf{u_{fluid}}})^T) \quad (4.3)$$


$$-\mathbf{n} \cdot \mathbf{q} = \rho C_p u_\tau \frac{T_w - T}{T^+} \qquad (4.4)$$

The fluid-flow geometry (Figure 20), is defined as a rectangle where the fluid enters to the right side with a speed of 25 m/s,
besides in the centre of the domain there is a piece of wall with two windows, covered by louvre shutters. Moreover two




types of louvre have been considered, one with an angle of 45° and another one of 60°. Finally the exit is on the left side,
where the hypothesized pressure is equal to 0 Pa. The hypotheses are:

- Incompressible fluid,

- Turbulent movement,

- Homogeneous density and speed profile.

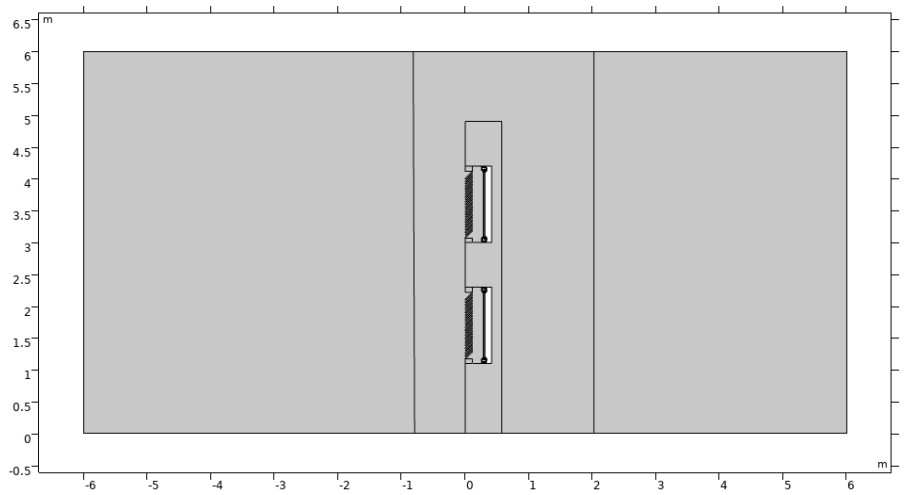


**Figure 20 Geometric model for CFD**

The physical characteristics of the fluid (Table 10) are:

| Table 10 Physical and mechanical properties of Tempered Glass ||
|---|---|
| Density | 2265 [kg/m$^3$] |
| Dynamic viscosity | 0.001 [Pa * s] |
| Thermal conductivity | 2.2 [W/m * K] |
| Heat capacity at constant pressure | 1255 [J/kg * K] |
| Ratio of specific heats | 1 |

Once the turbulent field had been studied, the solution obtained was used as initial values for the study of Heat Transfer in
Solids and Fluids, where the temperature is expressed through the function (3.7) through a time interval of 240 seconds.
The results highlight that shutters, commonly used, are not suitable for window protection against the flow. Indeed the
EPDM, in both cases considered, reaches the critical temperature after about 100 seconds. At the same time, polyamide
reaches its glass transition temperature after about 120 seconds (Fig. 21). Also, the glass is exposed to the effects of
pyroclastic flow, indeed it breaks due to thermal shock after 20 seconds (Fig. 22).





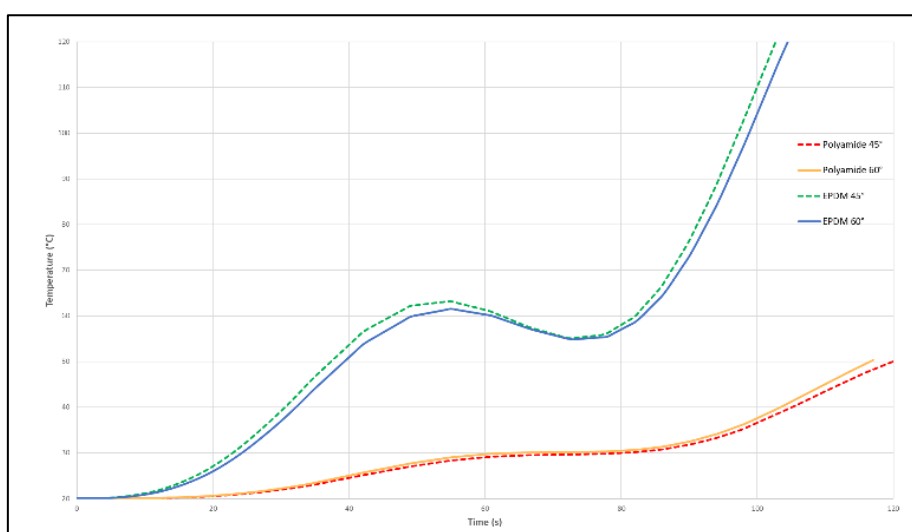

**Figure 21 Reached temperature by polyamide and EPDM**

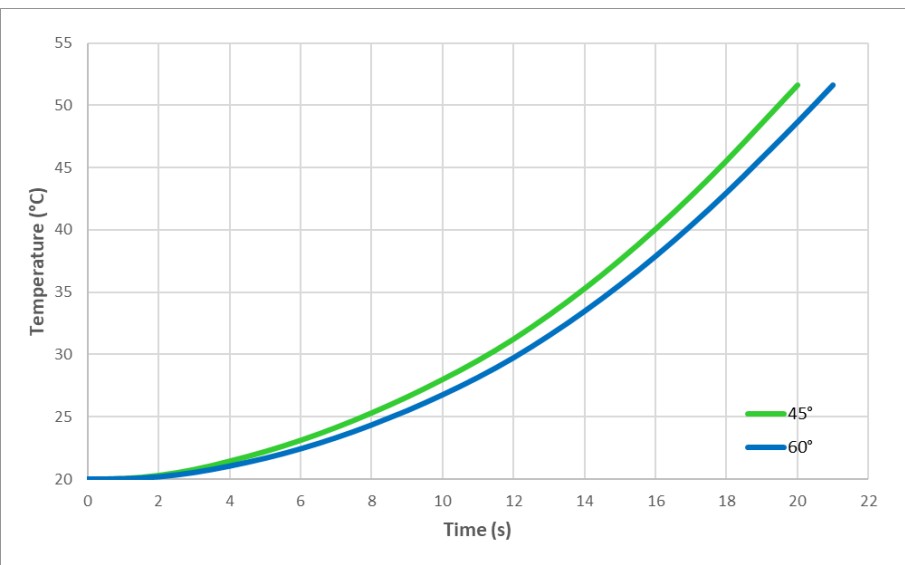

**Figure 22 Reached temperature by polyamide and EPDM**



## 370   7.New Panel for shutters

From the previous fluid-dynamic analyses it has emerged that a possible mitigation strategy against pyroclastic flow is the replacement of the louver system with a full panel shutter system (Fig.).

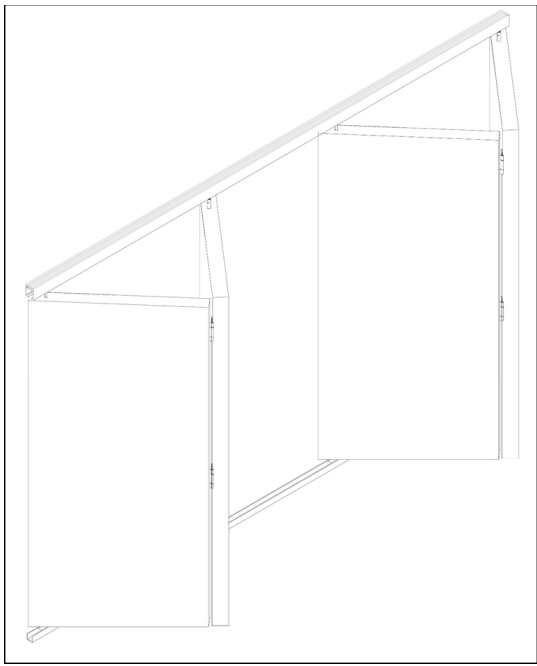

**Figure 23 Hypothesized shutter model**

The hypothesised panel is a stratified panel consisting of a 3 cm pine wood core placed between two 5 mm thick sheets of EN-AW 5005A aluminium alloy, whose characteristics are shown in Table 11, so the panel has a total thickness of 4 cm.

| Table 11 Physical and mechanical properties of Aluminium EN-AW 5005A | |
|---|---|
| Density | 2700 [kg/m$^3$] |
| Elastic Modulus | 70000 [MPa] |
| Ultimate Tensile Strength | 130 [MPa] |
| Poisson's ratio | 0.33 |
| Specific heat capacity | 900 [J/kgK] |
| Thermal conductivity | 238 [W/mK] |
| Thermal expansion | 3,7 e$^{-7}$ [1/K] |

This model was analysed to assess thermal stress in the 240-second time interval. So the geometrical model used is a panel
with a size (2.40x1.20) m equal to the size of a large opening; on the side surfaces a fixed constraint condition has been



placed (Fig.24) and on the external front a temperature expressed by the interpolated function (3.3) is placed which reaches a maximum temperature of 400°C.

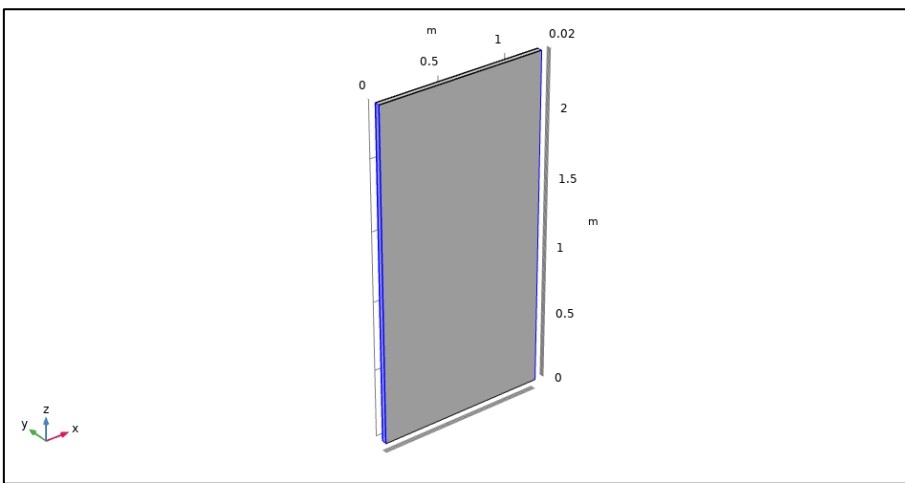

**Figure 24 Fixed constraints**

The first results highlight that the external aluminium sheet is resistant to temperature, reaching a maximum of 128 MPa (Fig.25). At the same time, the outer surface of the wooden core reaches a temperature of about 350 °C in 180 seconds (Fig. 26). According to Jenkins et al. (2009) when the wood reaches a temperature of 350 - 400 °C then it is possible to assume that the ignition probably takes place within three minutes; therefore it is necessary to evaluate the rate of carbonization of the wood considered. Eurocode 5 defines for solid wood with a characteristic density ≥ 290 kg/m3 a carbonization speed of

0.8 mm/min. In the case examined, therefore, carbonization will take place in the remaining 60 seconds, and it is, hence, possible to state that the wood panel will be intact as a result of the phenomenon considered. As a result of these analyses, the hypothesis of a new panel for shutters is suitable for the temperatures expected in the red areas.



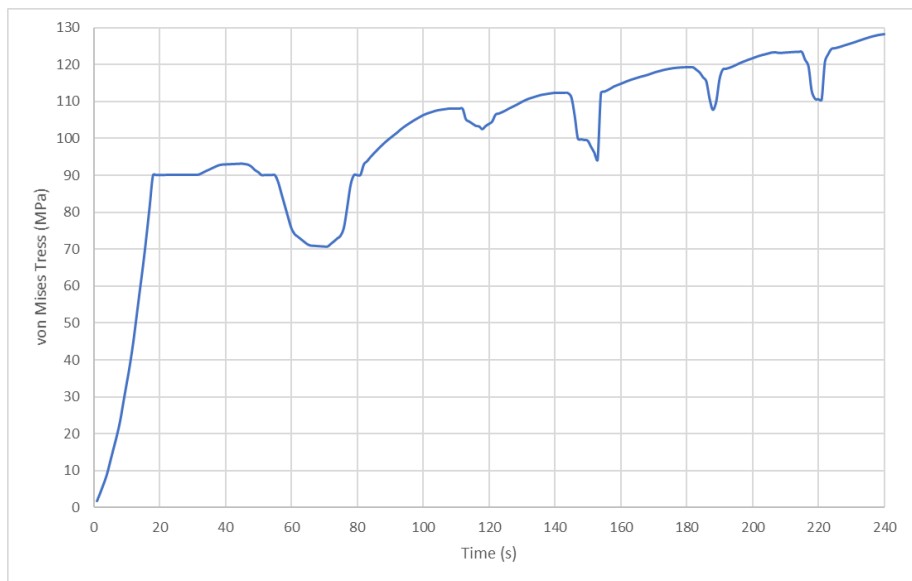

**Figure 25 von Mises stress of aluminium**

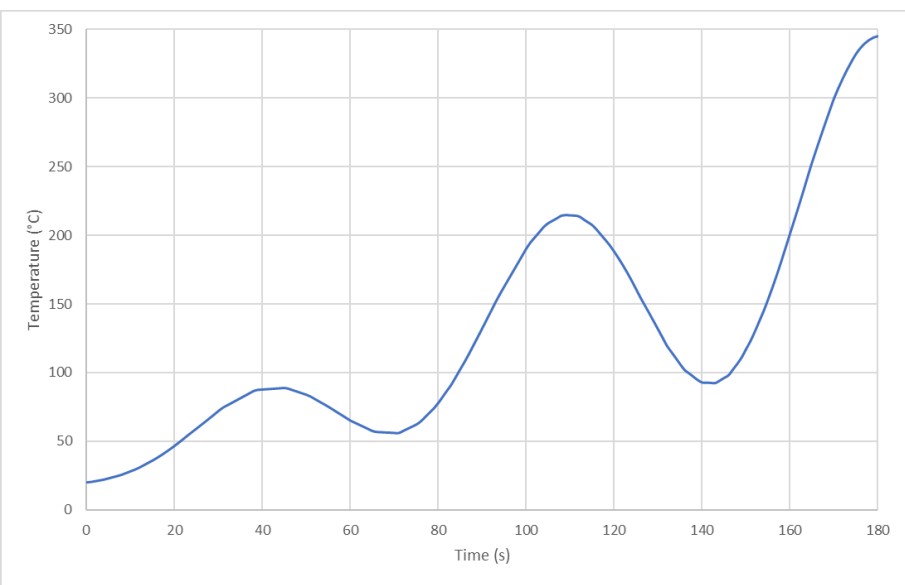


**Figure 26 Reached temperature of wood core**



## Conclusion

Research into mitigation technologies to reduce the volcanic risk of buildings represents a fundamental field of investigation for the definition of new resilient areas where rehabilitation plans will be faster and cheaper. Through this study, it has been demonstrated that incremental strategies represent a suitable strategy for the protection of the most vulnerable elements of the building. Moreover, these first analyses have shown that the pyroclastic flow represents a danger especially for the temperatures that characterise them, in fact not only glass fails as a result of thermal stress but also the other constituent

elements such as gaskets and thermal break are vulnerable as thermoplastics. Furthermore, fluid-dynamic analyses have shown that shutters, which had been hypothesised as a possible filter/protection element against flows, are unsuitable for this purpose. Therefore, it is necessary to provide for replacement with a sandwich panel, which can withstand the expected temperatures.

In the case of Vesuvius and Campi Flegrei, the principal impediment to the implementation of these proposals is the size of

the likely afflicted regions, which suggests relevant questions about the financial sustainability of mitigation measures. Local governments and residents need to be fully conscious of the likely damage resulting from an eruption and the cost-effectiveness of possible mitigation interventions. Regarding the economic, political and social "weight" of volcanic risk in densely populated areas, a valid method of assessing the effectiveness of mitigation solutions can contribute objective assistance to strategic choices and emergency plans. The approach discussed in this document, although regarding the area of

Vesuvius and Campi Flegrei in the specific parameters analysed (hazard characterisation, building typologies, construction technologies), represents a methodology that could also be adopted in other contexts. Further work is underway to understand the behaviour of shutters considering a fluid-dynamic problem and a 3D model of the window to analyse the contribution of further elements such as the hinges and the actual closing system, in addition, models for combined pressure and temperature analysis are being implemented.

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
