# Peer review of "Pyroclastic flow mitigation strategies: a new perspective for the red area"

_Natural Hazards and Earth System Sciences, 2020_

## Referee Comment (RC1) · Vittorio Bosi (Referee) · 16 Mar 2021

I'm glad I reviewed this article because it is consistent with what I do daily in my job. The vulnerability of structures to natural phenomena is in fact too often forgotten, to leave space to the hazard. In this article the Authors carefully analyse openings such as windows in case of volcanic eruption and the impact of pyroclastic flows, giving at the end some indications on possible mitigation strategies. They are giving a good contribution to society that can take their insights as a starting point to elaborate sustainable mitigation strategy. While fully understanding the purpose of the paper, it is unfortunate that the Authors do not go as far as cost/benefit considerations and energy performance improvement possibilities in applying such mitigation strategies. I

hope this is a next step, perhaps for a new paper

Please also note the supplement to this comment:
https://nhess.copernicus.org/preprints/nhess-2020-320/nhess-2020-320-RC1-supplement.pdf

**Supplement:**

Referee: Vittorio Bosi Department of Civil Protection of Italy Volcanic risk service

**COMMENT ON PAPER:**

**Pyroclastic flow mitigation strategies: a new perspective for the red area**

Authors: Mauro Iacuaniello, Andrea Montanino, Daniela De Gregorio, and Giulio Zuccaro

**Scientific significance**

The paper rises conclusion useful for the government institutions which can find in it some interesting indications about mitigations strategies. It gives a noticeable contribution to the understanding of how to defend our property to pyroclastic flow.

Moreover, the mitigation of risk from pyroclastic flows in an area such as the Phlegraean Fields is strongly influenced by general cost/benefit analysis, which is, in fact, very interesting to analyse. Taking into account the timing of eruptions in the Phlegraean Fields, the costs to be incurred for risk mitigation taking into account the suggestions given in this article, and the durability of windows and shutters, in relation to what could possibly be destroyed or preserved, including houses and goods inside, would maybe highlight the sustainability of those measures.

This, however, is a general comment that could change radically if applied to the current conditions of the volcano. The state of activity of the Phlegraean Fields today is in fact represented by the Yellow alert level and an operational phase of attention, with parameters that have been growing steadily for several years, leading to the acceleration of all volcanic risk mitigation policies, while the national emergency plan and sector plans are updated.

I believe that national and local governments should give due consideration to results such as those presented in this article, in order to create a real long-term, sustainable and simultaneously risk-mitigating business plan.

**Scientific quality**

Even though modelling is not my area of expertise, the paper is well written, with robust methodology. Obviously, there are several uncertainties and assumptions, due to make the modelling easier, but which in the end partially limit the final results. Nevertheless, the article is well written, one sentence was found in Italian, evidently not translated, and this is not synonymous with the utmost attention. The same occurred for two figures /fig. xx and fig xxx, without any number. The text is sometimes not very fluent due to incidental propositions that are not effective for easy reading. I suggest eliminating many of these

incidental propositions as indicated in the text. I suggest also to limit the use of semicolon or to use in a slightly different way.

Furthermore, I noticed that many concepts are just mentioned instead to be described assuming a good knowledge of the reader on the treated topic. I encourage the Authors to develop some of the relevant concepts (e.g. why a certain function is used instead of some others).

**Presentation quality**

Scientific data are well presented and explained. The paper is well structured, with correct paragraphs and references. In my opinion, there is a slight lack of explanatory charts, which would make reading easier and the results more understandable.

Figures could be a little more attractive, considering that, since this is an article that has a lot of modeling data, they are pretty similar.

The conclusions are consistent and well written, but could be clearer. A reader who is not particularly advanced would find it difficult to fully understand them.

If the authors would like, an analysis of whether and to what extent the proposed mitigation measures could also be valid for the energy efficiency of buildings could be very interesting.

**In the text:**

Line 25 - It would be better to avoid or limit the incisions in the writing to make the writing clearer, when it is possible.

Line 26 – Three thousand people... are you sure.. are they maybe 3 million people?

Line 28 – Floods and mudflows? Are you referring to "lahars"? If yes, it would be better to write (floods and mudflows, namely "lahars")

Line 31 – The main objective. Maybe "one of the main objectives?"

Line 45-48 – This sentences should be improved by removing a few semicolon.

Line 49-50 – this sentence is too long. It can be improved by a dot after the word "large" (see in the text)

Line 69 - After the word "inside" I would avoid "and". See in text

Line 84 – Fig xx???

Line 166-167 – Are (TG) and (Tg) the same glass transition temperature?

Line 181 – After a semicolon avoid the word "and". You do not need.

Line 254 – (fig. xxx) ???

Line 259 – There is a sentence in Italian "è stato necessario valutare lo stress termico (xxx). Translate please.

Line 261 – Add "s" in the word function (two functions).

Line 261 – There is the possibility to use other functions? If yes please explain why are you using the 3.3 and 3.4.

Line 410 – I think it is fair to add that another unfavourable condition is the interval between eruptions, which can be hundreds of years. Mitigation measures in a context such as that of the Phlegraean Fields and Vesuvius unfortunately have a very difficult financial sustainability in these times of recurrence, especially considering that no government has so far decided whether the people who would be removed could return to their homes or could be resettled elsewhere.

[revised manuscript text omitted]

---

## Author Comment (AC1) · 22 Mar 2021

Line 25 – It would be better to avoid or limit the incisions in the writing to make the writing clearer, when it is possible. Fixed

Line 26 – Three thousand people. . . are you sure.. are they maybe 3 million people? Fixed

Line 28 – Floods and mudflows? Are you referring to "lahars"? If yes, it would be better to write (floods and mudflows, namely "lahars") Fixed

Line 31 – The main objective. Maybe "one of the main objectives?" Yes indeed. Fixed

Line 45-48 – This sentences should be improved by removing a few semicolon. Fixed

Line 49-50 – this sentence is too long. It can be improved by a dot after the word "large" (see in the text) Fixed

Line 69 – After the word "inside" I would avoid "and". See in text Fixed

Line 84 – Fig xx??? Fixed

Line 166-167 – Are (TG) and (Tg) the same glass transition temperature? Yes they are the same, and I fixed the typo

Line 181 – After a semicolon avoid the word "and". You do not need. Fixed

Line 254 – (fig. xxx) ??? Fixed

Line 259 – There is a sentence in Italian "è stato necessario valutare lo stress termico (xxx). Translate please. Fixed

Line 261 – Add "s" in the word function (two functions). Fixed

Line 261 – There is the possibility to use other functions? If yes please explain why are you using the 3.3 and 3.4. Yes, I am studying others, but in this paper I have just showed the results of these first functions.

Line 410 – I think it is fair to add that another unfavourable condition is the interval between eruptions, which can be hundreds of years. Mitigation measures in a context such as that of the Phlegraean Fields and Vesuvius unfortunately have a very difficult financial sustainability in these times of recurrence, especially considering that no government has so far decided whether the people who would be removed could return to their homes or could be resettled elsewhere. That is true, indeed I have specified it in the new conclusion. Besides I am studying new mitigation strategy which can also represent an energy-saving strategy.

---

## Referee Comment (RC2) · Anonymous Referee #2 · 3 Jun 2021

The bibliography refers mainly to the authors' publications. It is necessary to complete it with various references.

The objective of this research paper should be clearly exposed at the beginning.

A section on the research method should also be included. The method must be detailed and argued

---

## Author Comment (AC2) · 5 Jul 2021

•"The objective of this research paper should be clearly exposed at the beginning."

Through the proposed mitigation strategy, the task is to establish a resilient sector within the red areas where buildings can withstand pressure and temperature to achieve both economic benefits and decrease the time of reconstruction.

•"A section on the research method should also be included. The method must be detailed and argued" The document sets out the methodology for both the exposure assessment and for the vulnerability assessment.

• The bibliography refers mainly to the authors' publications. It is necessary to complete it with various references

Unfortunately, there is not an extensive bibliography about the volcanic risk and mitigation strategy. Nevertheless, some new pieces of research have been referred to; such as • Todesco, M., Neri, A., Ongaro, T. E., Papale, P., Macedonio, G., Santacroce, R.,and Longo, A. (2002). Pyroclastic flows hazard assessment at Vesuvius (Italy) by using numerical modeling. i. large-scale dynamics. Bulletin of Volcanology, 64(3):155–177. • Walker, G. P. (1973). Explosive volcanic eruptions—a new classification scheme. Geologische Rundschau, 62(2):431–446. • Newhall, C., Self, S., and Robock, A. (2018). Anticipating future Volcanic Explosivity Index (VEI) 7 eruptions and their chilling impacts. Geosphere, 14(2):572–603. • Newhall, C. G. and Self, S. (1982). The volcanic explosivity index (vei) an estimate of explosive magnitude for historical volcanism. Journal of Geophysical Research: Oceans, 87(C2):1231–1238. • Ongaro, T. E., Komorowski, J.-C., Legendre, Y., and Neri, A. (2020). Modelling pyroclastic density currents from a subplinian eruption at la soufrière de Guadeloupe (west indies, france). Bulletin of Volcanology, 82(12):1–26.